# Porcine Circovirus Type 2 Vaccines: Commercial Application and Research Advances

**DOI:** 10.3390/v14092005

**Published:** 2022-09-10

**Authors:** Jinshuo Guo, Lei Hou, Jianwei Zhou, Dedong Wang, Yongqiu Cui, Xufei Feng, Jue Liu

**Affiliations:** 1Department of Preventive Veterinary Medicine, College of Veterinary Medicine, Yangzhou University, Yangzhou 225012, China; 2Jiangsu Co-Innovation Center for Prevention and Control of Important Animal Infectious Diseases and Zoonoses, Yangzhou University, Yangzhou 225012, China

**Keywords:** PCV2, PCVAD, commercial vaccines, experimental vaccines, protective efficiency, adjuvant

## Abstract

Porcine circovirus type 2 (PCV2) infection can lead to porcine circovirus-associated disease (PCVAD), causing great economic losses to the global swine industry. Conventional vaccination programs are a major measure in the prevention and control of this disease. Currently, there are 5 commercially available PCV2 vaccines in the international market and 10 kinds commercially available PCV2 vaccines in the Chinese market that confer good efficacy against this virus by alleviating clinicopathological manifestations and enhancing growth performance in pigs. In addition, diverse experimental PCV2 vaccines with protective efficiency have been developed, including attenuated chimeric, nucleic acid, subunit, multivalent, and viral-vectored vaccines. These experimental vaccines have been shown to be relatively effective in improving the efficiency of pig production and simplifying prevention procedures. Adjuvants can be used to promote vaccines with higher protective immunity. Herein, we review the application of multiple commercial vaccines over the years and research advances in experimental vaccines, which provide the possibility for the development of superior vaccines to successfully prevent and control PCV2 infection in the future.

## 1. Introduction

Swine infectious diseases profoundly impact the livestock breeding industry, international trade, and human public health. Porcine circovirus type 2 (PCV2)-associated disease (PCVAD) was first identified in 1991 in pigs with signs of postweaning multisystemic wasting syndrome (PMWS), now called PCV2-systemic disease [1,2]. In addition, PCV2 is associated with subclinical infection, necrotizing pneumonia, encephalopathy, congenital tremor, and reproductive disorders [3,4,5]. Infection with PCV2 induces immunosuppression in pigs, thereby leading to enhanced susceptibility to other infectious agents and a poor immune response to vaccination. PCVAD is now recognized as one of the most important diseases in pig-rearing countries and regions and has caused severe economic losses to global pig production [6].

PCV2, a member of the genus *Circovirus* within the family *Circoviridae*, is one of the smallest DNA viruses, with a diameter of 16–18 nm [7]. The genome of PCV2 is a small, single-stranded, closed-circular, 1.7 kb long structure [8]. Approximately 10 open reading frames (ORFs) were detected, mainly consisting of two stem-loop-connected head-to-head, namely, ORF1 and ORF2 [9]. ORF1 encodes a 35.7 kDa replicase protein involved in viral replication [10]; ORF2 encodes a 27.8 kDa structural capsid (Cap) protein, which functions primarily as a host-protective immunogen [11,12]. ORF3 protein induces an apoptotic response and contributes to viral pathogenesis in vitro and in vivo [13,14,15], whereas ORF4 protein inhibits caspase activity and suppresses the proliferation of CD4^+^ and CD8^+^ T cells [16].

PCV2 consists of eight genotypes, PCV2a to PCV2h, with the main genotypes being PCV2a, PCV2b, and PCV2d, whereas PCV2c, PCV2e, PCV2f, PCV2g, and PCV2h are the less prevalent genotypes [17,18,19,20,21,22,23]. A retrospective study revealed that PCV2a was first detected in Germany in 1962 [24], and was the most prevalent strain in the following decades. PCV2 has the highest nucleotide substitution rate (1.2 × 10^−3^ substitutions/site/year) among the known single-stranded DNA viruses [18,25,26]. PCV2 gradually underwent a genotypic shift, with epidemic strains worldwide slowly shifting from PCV2a to PCV2b before the first commercial vaccines became available in 2006, and the virus became considerably more virulent [17,27,28,29]. In recent years, PCV2 has undergone a second genotypic shift, with the main prevalent strain, PCV2b, changing to PCV2d [17,18,30]. The emerging PCV2d strain has been found in an increasing number of vaccinated pigs worldwide, and it is speculated that PCV2d may break through the protection provided by existing vaccines [17,18,30,31,32,33]. Similar to the global PCV2 epidemic situation, in China, PCV2b was the main epidemic genotype from 2002 to 2008, but after 2009, PCV2d gradually became an epidemic genotype. In addition to PCV2a, PCV2b, and PCV2d genotypes, PCV2f and PCV2h genotypes were also identified in China [34,35,36]. Currently, the main method of PCV2 prevention and control is vaccination. Early commercial vaccines for PCV2 infection were mainly prepared with PCV2a and enhanced growth performance in pigs at both individual and population scales; however, PCV2b and PCV2d pose new challenges for the development of PCV2 vaccines. This paper reviews the progress in commercial and experimental vaccine research on PCV2 with the aim of providing theoretical references for PCV2 prevention, control, and vaccine development.

## 2. Commercial PCV2 Vaccines

Commercialized vaccines for PCV2 are classified into two main categories: inactivated vaccines and subunit vaccines that use Cap as the immunogen. Currently, all five kinds of PCV2 vaccines on the international market have been reported to significantly reduce clinicopathological manifestations and improve growth performance in PMWS-affected pigs [37]. In addition, vaccinated pigs have increased average daily gain and percentage of lean meat production, improved feed conversion, and reduced medication costs for vaccinated herds [38,39].

### 2.1. PCV2 Inactivated Vaccine

PCV2-infected cells are inactivated by physical or chemical (formalin) methods, so that PCV2 loses the ability to infect cells, but still maintains good immunogenicity. An adjuvant was added for emulsification to prepare the PCV2 inactivated vaccine. In 2006, Circovac^®^, the world’s first PCV2 vaccine with light paraffin oil as an adjuvant, was successfully developed by Merial (France) and marketed after EU approval. The vaccine was approved for the vaccination of sows (2 mL) and piglets (0.5 mL) [40]. The vaccine effectively reduced the incidence of viremia, viral tissue load, shedding, and transmission and promoted the growth of newborn piglets [41,42]. Another study showed that sperm quantity and quality did not alter after vaccination [43]. Subsequently, researchers discovered that Circovac^®^ could cross-protect pigs against PCV2b and PCV2d challenges, offering the possibility of protection against more PCV2 genotypes [41,44]. The second inactivated vaccine is Fostera^TM^ PCV, an inactivated PCV1-2a chimeric vaccine produced by Pfizer Animal Health, Inc. USA. In this vaccine, the ORF2 gene of PCV2a is cloned into the genome backbone of apathogenic PCV1, and Fostera^TM^ PCV contains sulfolio-cyclodextrin in squalane-in-water as an adjuvant [37,45,46,47]. The results showed that Fostera^TM^ PCV produced high levels of anti-PCV neutralizing antibodies very quickly after vaccination, significantly reducing the duration and concentration of PCV2 viremia when compared with that in the PCV2 alone-inoculated control group [48]. However, other studies showed that vaccination with Fostera^TM^ PCV did not produce high levels of neutralizing antibodies [49] and could not prevent vertical transmission [50]. Inactivated vaccines do not stimulate cellular immunity and often require adjuvants to enhance their effects.

### 2.2. Subunit Vaccine

A virus subunit vaccine is made from the components of the main immunogen of the virus using genetic engineering. Commercial PCV2 subunit vaccines have been designed and produced, mainly based on ORF2. Cap protein has neutralizing epitopes; therefore, it has the potential to induce protective immune responses [51,52]. Currently, three subunit vaccines (Circoflex^TM^, Circumvent, and Porcilis PCV^®^) are licensed for use in the international market, and all use the expression of ORF2 in a baculovirus system. After inactivation and purification, virus-like particles (VLPs) become morphologically identical to PCV2 particles [44]. The vaccine was able to elicit a strong PCV2-specific neutralizing antibodies and prevent the development of PCV2 viremia, with considerably decreased nasal and faecal viral shedding after different PCV2 isolates [44]. In a study evaluating the efficacy of Porcilis PCV^®^ vaccination against challenge with different PCV2 isolates, only one dose of Porcilis PCV^®^ significantly elicited cell-mediated immunity and reduced PCV2 viremia [53]. Another experiment simultaneously evaluated four commercial PCV2 vaccines; 3-week-old piglets were vaccinated with Circovac^®^, Fostera^TM^ PCV, Circoflex^TM^, and Porcilis PCV^®^ for four weeks and then challenged with PCV2. The average daily weight gains in all four vaccinated group were higher than those in the PCV2 alone-inoculated group at 14 and 21 days after challenge; however, compared with the two commercial inactivated vaccines, the subunit vaccines Circoflex^TM^ and Porcilis PCV^®^ showed lower virus load and shedding, milder histopathological lesions, lower scores of PCV2-antigens in lymph nodes, and higher levels of PCV2-specific neutralizing antibodies and cellular mediated immunity [48].

### 2.3. PCV2 Vaccines Marketed in China

Porcine circovirus infection in pig herds was first reported in China in 2000, and subsequently spread to many provinces in China [34]. Infection of PCV2 reduces the immunity of the pigs and affects the growth performance of pigs, which has caused huge losses to the Chinese pig industry. The Harbin Veterinary Research Institute of the Chinese Academy of Agricultural Sciences developed the first PCV2 (strain LG) inactivated vaccine in China. Subsequently, about 10 kinds PCV2 vaccines, including inactivated, subunit, and vectored vaccines, have been developed and commercially applied in the Chinese market (Appendix A).

## 3. Experimental PCV2 Vaccines

In addition to the commercial vaccines mentioned above, scholars have extensively studied various types of experimental PCV2 vaccines, such as inactivated, attenuated, live-attenuated, DNA, subunit, and multivalent vaccines. These experimental vaccines are expected to address the challenges of the PCV2 genotype transition and are beneficial for simplifying the tedious vaccination procedure.

### 3.1. Attenuated Chimeric Vaccines

Chimeric vaccines modify the structure of pathogens at the genetic level, splice or replace the genome fragments of two or more pathogens into one vector, and construct a recombinant vector that can express a variety of antigenic substances that can be used to prepare attenuated and inactivated vaccines. The currently reported experimental live-attenuated PCV2 vaccines were prepared from chimeric PCV1-2a and PCV1-2b strains. Studies on the live-attenuated chimeric PCV1-2a vaccines showed that the attenuated chimeric PCV1-2a was genetically stable in cell culture, serially passaged in pigs [54], and not pathogenic under experimental conditions [50]. Chimeric viruses PCV1-2a can induce protection of pigs against challenge with PCV2b or PCV2d. The virulence of the chimeric virus PCV1-2a was significantly weakened, and no fatal cases were found when administered to pigs. The extent of histopathological damage and the number of PCV2-specific antigens in the lymphoid tissue were consistently lower in pigs than in wild-type PCV2b-infected pigs. Pigs inoculated with attenuated chimeric PCV1-2b were challenged with wild-type PCV1-2a and PCV1-2b, the viral load in the pig serum and tissue was significantly reduced, and the clinical symptoms were significantly alleviated, indicating that the attenuated chimeric PCV1-2b virus elicited strong humoral immunity and induced a cross-protective immune response against PCV2a and PCV2b challenge in pigs [55]. In addition, PCV1-2b induces numerous IFN-γ-SCs [56,57]. PCV1-2b may be candidates for improved live attenuated vaccines. Notably, chimeric PCV1-2a was isolated from an acute outbreak of porcine reproductive and respiratory syndrome (PRRS) in Canada in 2008 [58]. Therefore, it is speculated that there are two possibilities for the origin of the PCV1-2 chimeric strain—one is from the artificial strain of the chimeric vaccine, and the other is due to the natural occurrence of PCV1 and PCV2 recombination [58].

### 3.2. Nucleic Acid Vaccines

Nucleic acid vaccines use genetic engineering to construct recombinant vectors and gene injection technology for immunization. As a new type of vaccine, the immune mechanism of the body can be activated in a specific manner to achieve rapid immunity. Nucleic acid vaccines are mainly divided into DNA and RNA vaccines, which play an important role in the prevention and control of virus infections. The current research on PCV2 nucleic acid vaccines is focused on DNA vaccines. Vaccination methods for nucleic acid vaccines vary, and choosing an appropriate nucleic acid vaccination method will help improve the efficacy of the vaccine. In 2004, Kamstrup et al. took the lead in researching a DNA vaccine against PCV2 and immunized mice thrice with a gene gun; the PCV2 DNA vaccine could increase the level of PCV2 antibodies in mouse serum [59]. Zhang et al. also found that a DNA vaccine of PCV-like virus P1 can stimulate milder PCV2 viremia and histopathological changes after two intramuscular immunizations in mice as compared with the non-vaccinated group after PCV2 challenge [60]. After immunizing mice with the DNA vaccine based on the PCV2 *Cap* gene, high levels of highly specific serum antibodies and cytokines (interferon-γ and interleukin (IL)-10) were induced, and the PCV2 viral load in tissues was reduced, conferring strong protection against PCV2 infection [61]. However, DNA vaccines are limited by issues such as plasmid degradation and delivery. To overcome these limitations, researchers have explored the enhancement of DNA immunization-induced immune responses by optimizing plasmid construction, adjuvant selection, and vaccine delivery systems [62]. In the future, nucleic acid vaccines may become alternative biological products in the fields of animal husbandry disease prevention and control and human disease prevention and control, and will have broad application prospects.

### 3.3. Virus-like Particles (VLPs) Subunit Vaccines

VLPs mimic the structure of a virus. VLP vaccines can stimulate strong B and T lymphocyte-mediated responses. Recombinant Cap proteins expressed in insect cells can form VLPs for the development of commercial PCV2 subunit vaccines [51]. However, owing to the limitations of the cell culture cost and purification time, it is still necessary to improve this method. The N-terminus of the Cap protein has a nuclear localization signal (NLS) [63], and it is rich in arginine residues and contains several rare codons that hinder the expression of foreign genes in *E. coli*. The expression level of the Cap protein can be increased by codon optimization and by using the *E. coli* host strain containing an extra copy of the rare tRNA gene. Wu et al. successfully expressed full-length Cap1-233 VLP without any fusion tag in the *E. coli* system and found that Cys108 is necessary for capsid assembly [64]. PCV2 VLP-immunized pigs can induce specific antibody responses and are resistant to PCV2 challenge [64], which provides the basis for the large-scale production of PCV2 VLP vaccines. Chi et al. used gE-deficient pseudorabies virus (PRV) to construct a recombinant gE(−)/PCV2 cap(+)PRV by replacing the upstream part of the gE gene with the *Cap* gene of PCV2, in which the expressed Cap protein self-assembled into VLPs and the purified VLPs could induce a significant immune response to PCV2 in immunized mice or guinea pigs [65].

The phage small capsid protein D-fusion polypeptide can be expressed by plasmids in *E. coli* and remains soluble to form phage particles with exogenous polypeptides for vaccine production. Some researchers have constructed PCV2 Cap into phage lambda particles and expressed it in *E. coli*, which induced PCV2 neutralizing antibodies and elicited B and T cell-mediated immune responses in pigs [66]. The hydrophobic region of tropoelastin has elastin-like peptides (ELPs), and the repeat sequence motif is valine-proline-glycine-Xaa-glycine, in which Xaa may be any amino acid except proline. In *E. coli*, ELP was expressed as a fusion to the PCV2 Cap protein, which was purified to a high purity using an inverse transition cycle. The purified ELP-Cap fusion protein was assembled into Cap protein VLPs (ELP-VLPs). After comparing the efficacy of the ELPylated VLP PCV2 vaccine with the commercial inactivated Yuan li jia vaccine and VLP-based CircoflexTM vaccine, both virus neutralizing antibodies and interferon responses were significantly higher than those of the two commercial vaccines [67].

### 3.4. Viral Vectored Vaccines

In the production process of the breeding industry, swine often needs to be vaccinated against multiple pathogens. Bivalent and multi-component vaccines will dramatically simplify the vaccination plan and reduce the economic burden on the pig industry. To increase the width of the vaccine epitope, a PCV2a–PCV2b bivalent vaccine was developed. The cross-protection of the PCV2 vaccine against different genotypes was not satisfied, but the bivalent PCV2 vaccine provided superior protection [68]. The porcine IL-18 gene, PCV2 Cap, and M-like protein (SzP) gene of *Streptococcus equi* ssp. zooepidemicus (SEZ) were inserted into the swine pox virus (SPV) genome to construct the recombinant swinepox virus rSPV-ICS, in which the rSPV-ICS conferred good protection against co-challenge of PCV2 and SEZ [69]. The combined vaccine based on porcine PCV2 Cap VLP and porcine parvovirus (PPV) VP2 VLP significantly improved the growth performances of piglets and could alleviate PCVAD and related diseases caused by PCV2 and PPV co-infection [70]. A recombinant BacSC-Dual-GP5-Cap vaccine, which simultaneously expressed PRRSV GP5 glycoprotein and PCV2 capsid protein in the baculovirus system, can elicit considerable levels of virus neutralization titers and induce lymphocyte proliferation responses [71]. PCV2 and *Mycoplasma hyopneumoniae* bivalent vaccines (Fostera^TM^ PCV) also have good protections [72]. Adenoviral expression vectors have the advantage of a simple construction and induction of humoral, mucosal, and cellular immunity [73,74]. Adenovirus-vectored vaccines of granulocyte-macrophage colony-stimulating factor (GMCSF) and CD40L or intron A and WPRE-modified PCV2 Cap are not as effective as the commercial inactivated PCV2 vaccine. The reconstructed recombinant adenovirus vaccine Ad-A-spCD40L-spCap-spGMCSF-W was combined with the commercial inactivated vaccine PCV2 SH-strain. The recombinant adenovirus vaccine demonstrates a potent immune response and superior protection and is expected to become a candidate vaccine for PCVAD [75]. In addition, studies have shown that the *Yersinia pseudotuberculosis* invasin (Inv) protein can enhance the immune response and can be used to construct a recombinant adenovirus containing the PCV2b Cap gene and InvC gene, which has a good immune response [76].

## 4. Application of Adjuvants

Adjuvants can enhance the pig’s immune response to antigens, alter the type of immune response, enhance antibody levels, or generate more effective protective immunity. Adjuvants for PCV2 vaccines are mainly divided into two categories: chemical and molecular adjuvants (Table 1).

### 4.1. Chemical Adjuvants

Inactivated PCV2 whole-virus vaccines with oil adjuvants can elicit high levels of humoral immune responses in pigs [77]. After inoculation with DNA PCV2 vaccine comprising liposome adjuvant, pigs had significantly enhanced levels of interferon-γ-secreting cells and neutralizing antibodies, whereas PCV2 viremia levels were significantly reduced after PCV2 challenge [78].

### 4.2. Molecular Adjuvants

The peptide binding C-terminal portion of mycobacterium tuberculosis heat shock protein 70 (HSP70) exerts an adjuvant effect when fused to PCV2 Cap gene to construct a PCV2 DNA vaccine, which exhibits enhanced serum immunoglobulin G levels and T helper 1 immune responses in mice [79]. Moreover, PCV2 DNA vaccine conjugated molecular adjuvant ubiquitin showed an increased Th1 type cellular immune response and Cap-specific antibody production in mice [80]. IL-2 and GMCSF, which were regarded as immune adjuvants, can effectively increase the protective effect of the PCV2 Cap-based subunit vaccine in mice [81]. Chitosan oligosaccharides (COS) has been used as an adjuvant to improve immunogenicity of PCV2 vaccines by physically mixing with inactivated vaccine or covalently linking to subunit vaccine [82,83]. COS with a high degree of deacetylation have similar properties to commercial adjuvants [83]. When further conjugated to a carrier protein (Ovalbumin), the COS-OVA-PCV2 vaccine showed higher levels of PCV2-specific antibody production and cytokine secretion than PCV2 vaccine physically mixed with a commercialized adjuvant ISA206 [84]. Porcine CD40 ligand (CD40L) and granulocyte-macrophage colony-stimulating factor (GMCSF) can synergistically enhance the humoral and cellular immune responses of PCV2 adenovirus vaccine (Ad-CD40L-Cap-GMCFS), as evidenced by enhanced PCV2 specific antibody titer and neutralizing activity and increased lymphocyte proliferation activity and Th1-type cytokine levels, as well as decreased virus loads in mice after PCV2 challenge when compared with that of Ad-Cap, Ad-CD40L-Cap, or Ad-Cap-GMCSF group [85]. Microbial ligands for TLRs include unmethylated cytosine phosphodiester guanine (CpG) motifs of bacterial DNA and bacterial flagellin, which are considered as pathogen-associated molecular patterns (PAMPs). It has been reported that CpG motifs have immunostimulatory effects on PCV2 as immune adjuvants [86]. The terminal degradation product (C3d) of the mammalian complement component C3 binds to complement receptor type 2 on B cells to regulate adaptive immune responses and is used as a vaccine adjuvant. The conjugation of multiple copies of the C3d molecule or its minimal binding domain C3d-P28 to the vaccine can greatly enhance vaccine-specific responses [93,94]. pVAX1-ORF2-C3d-P28.3 DNA vaccine, which was constructed using CD3-P28 and ORF2 gene of PCV2d, induced humoral and cellular immune responses to protect pigs against challenge of the PCV2b and PCV2d subtypes [87]. As an immune adjuvant, porcine IFN-γ substantially increased the protective immune response of the Cap protein-based subunit vaccine against PCV2 challenge when co-vaccinated to mice [88]. Flagellins of *Salmonella typhimurium* (FliC and FljB) exert immunomodulatory effects by activating TLR 5-positive DCs [95,96]. The fusion expression of flagellin and PCV2 Cap in the baculovirus system or the use of flagellins of FliC as adjuvants in PCV2 vaccines can enhance B and T cell-mediated immune responses in mice and pigs [89,90,91,92].

## 5. Other Issues with PCV2 Vaccines

### 5.1. Difficulty in the Development of Attenuated PCV2 Vaccine

PCV2 vaccine development has been in a state of continuous updating because of the high mutation rate of the PCV2 genome and the emergence of new PCV2 subtypes. Compared with inactivated vaccines, attenuated vaccines better mediate cellular immunity and produce faster immune effects. The Fenaux study found that, after 120 serial passages of the PCV2 strain in PK-15 cells, only 328 (C328G) and 573 (A573C) mutations occurred in ORF2 gene, and the resultant PCV2 showed lower virulence to SPF piglets [97]. However, potential virulence of the resultant PCV2 vaccine after 120 serial passages may pose a risk for the actual clinical application.

### 5.2. Selection of Cells Suitable for PCV2 Proliferation

PCV2 relies on proteins expressed in the S phase of mitosis during replication, and D-glucosamine was added to stimulate cell division during PK-15 cell culture, thereby promoting PCV2 replication [3]. Hirai et al. found that primary cells from pig liver are more suitable for PCV2 proliferation than pig kidney cells [98]. Qingdao Oland Better Bioengineering Co., Ltd. selected a cloned PK15-B1 cell line to enhance PCV2 titer for producing PCV2 vaccines, and the titer of PCV2 before inactivation is ≥10^7.5^ TCID_50_/mL. However, most of the inactivated PCV2 vaccines currently are still developed using conventional PK-15 cells.

### 5.3. Complexity of PCV2 Vaccine Evaluation

Determination of virus loads in tissues and blood, scoring of pathology and PCV2 antigen-distribution in lymph nodes, levels of PCV2-specific antibodies, activity of lymphocyte proliferation, levels of cytokine production, and/or growth performance are used to evaluate the efficacy of PCV2 vaccines [82,86]. However, there is currently no universal standard for the efficacy evaluation of PCV2 vaccines.

## 6. Conclusions

At present, commercial vaccines applied in the international and Chinese markets have good clinical effects in managing PCVAD and are effective against PCV2 infection. However, as PCV2 genotypes increase, developing vaccines that offer prevention and control is crucial. Studies have found that some experimental PCV2 vaccines can stimulate the body’s humoral and cellular immunity and are superior to commercially available vaccines in reducing PCV2 viremia and increasing serum antibody levels. Several studies have reported that the PCV2 VLP vaccine has a complete PCV2 viral capsid and is immunogenic. Therefore, the focus is mainly on the development of adjuvants to improve vaccine efficacy. In the prevention of human diseases, in vitro transcribed (IVT) mRNA has gradually attracted the attention of researchers because of its effectiveness, fast production speed, low cost, and ability to deal with viral mutations. However, (IVT) mRNA vaccines have not yet been used as candidates for experimental PCV2 vaccines, mainly because of relatively new technology and transportation constraints. With the resolution of these problems, mRNA vaccines can be used to prevent and control PCVAD. Furthermore, the development speed of PCV2 vaccines lags behind the evolution speed of PCV2; therefore, it is particularly important to study the epidemiology behind the variation in PCV2 epidemic strains.

## Figures and Tables

**Table 1 viruses-14-02005-t001:** Application of adjuvants in PCV2 experimental vaccines.

Type of Adjuvant	Adjuvant	Vaccine	References
Chemical adjuvant	Oil adjuvant	Inactivated vaccine	[77]
Chemical adjuvant	Liposome	DNA vaccine	[78]
Molecular adjuvant	HSP70	DNA vaccine	[79]
Molecular adjuvant	Ubiquitin	DNA vaccine	[80]
Molecular adjuvant	IL-2 and GMCSF	Subunit vaccine	[81]
Molecular adjuvant	COS	Inactivated/subunit vaccine	[82,83]
Molecular adjuvant	COS-OVA	Inactivated vaccine	[84]
Molecular adjuvant	CD40L and GMCSF	Adenovirus vaccine	[85]
Molecular adjuvant	CpG motifs	DNA vaccine	[86]
Molecular adjuvant	C3d-P28	DNA vaccine	[87]
Molecular adjuvant	IFN-γ	Subunit vaccine	[88]
Molecular adjuvant	Flagellin	Subunit vaccine	[89,90,91,92]

## Data Availability

Not applicable.

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
