# Peer review of "Porcine Circovirus Type 2 Vaccines: Commercial Application and Research Advances"

_viruses, 2022, doi:10.3390/v14092005_

Round 1

Reviewer 1 Report

In this manuscript, authors reviewed advances of multiple PCV2 vaccines, application of adjuvants, and the issue of vaccine research and development. In general, this review has an important reference for our reader. I suggest the editor considers the Manuscript for publication. However, I have seral concerns listed as below:

1. The citation and reference need to be standardized.

(1) Some sentences do not cite the literature, for examples:

Lines 29-31: Porcine circovirus type 2 (PCV2)-associated disease (PCVAD), was first identified, in 1991, in pigs with signs of postweaning multisystemic wasting syndrome (PMWS), now called PCV2-systemic disease.

Line 120-123: Porcine circovirus infection in pig herds was first reported in China in 2000, and subsequently spread to many provinces in China. Infection of PCV2 reduces the immunity of the pigs and affects the growth performance of pigs in which has caused huge losses to the Chinese pig industry.

Lines 190-191: Wu et al. successfully expressed full-length Cap1-233 VLP without any fusion tag in the E. coli system and found that Cys108 is necessary for capsid assembly.

(2) A literature was lost in the list of reference:

Line 152-156: A literature cited in manuscript was lost in the list of reference: Notably, chimeric PCV1-2a was isolated from an acute outbreak of porcine reproductive and respiratory syndrome (PRRS) in Canada in 2008 [48].

2. The efficacy evaluation of the 10 PCV2 vaccines in Chinese market was not described in this review, so authors need add some relevant experimental reports in the MS. Otherwise, the content of "2.3. PCV2 Vaccines Marketed in China" may be considered to be removed. Besides, the "Table S1" could not be found in this manuscript.

3. Currently, the present study demonstrated that PCV2 can be divided into 8 genotypes (PCV2a-PCV2h), please double check.

Ref: Franzo G, Segalés J. Porcine circovirus 2 (PCV-2) genotype update and proposal of a new genotyping methodology. PLoS One. 2018 Dec 6;13(12):e0208585

4. The formatting of the text needs to be adjusted as follows:

Line 38: "Circovirus" and "Circoviridae" should be italic.

Line 112: Porcilis PCV® in place of Porcilis PCV®

Line 209: "commercial inactivated vaccine Yuan li jia"- should be changed to-"commercial inactivated Yuanlijia vaccine"

5. The manuscript should be edited by a native English speaker who is familiar with scientific writing.

Reviewer 2 Report

Dear authors

I hope all of you are fine. Regarding the revision of the Manuscript ID (viruses-1908339), titled “Porcine Circovirus Type 2 Vaccines: Commercial Application and Research Advances”. This is a very important and valuable review article indicating the situation of porcine circovirus type 2 vaccines in China. However, some comments should be replied for better improvement of this review. 

Comments should be replied.

  1. I think authors should mention in the introduction (lines 56-60) a small hent to the history, current situation and economical importance of PCV2 as well as the detected genotypes in China.
  2. Line 140: delete this sentence, prepare live attenuated chimeric PCV2 vaccines, or rewrite it correctly.
  3. Line 177-180: (In the future, nucleic acid vaccines will become indispensable biological products in the fields of animal husbandry disease prevention and control and human disease prevention and control and will have broad application prospects) is it your expectation or based on scientific research, please add reference(s).
  4. Line 181: VLP Subunit Vaccines should be corrected to Virus Like Particles (VLP) Subunit Vaccines.
  5. Line 212-241: I think it is better to merge both paragraphs under the same title (Viral vectored vaccines).
  6. Line 273: please write the full name of CpG as Cytosine phosphodiester Guanine (CpG).
  7. Please mention that CpG and Flagellin are considered as Pathogen Associated Molecular Patterns (PAMP).
  8. Finally in conclusion, lines 325-328: as your advice here to develop mRNA vaccines to protect against PCV2, it is more accurate to specify these vaccines as an in vitro transcribed (IVT) mRNA or plasmid DNA (pDNA) as you talking about a DNA virus (PCVAD).
